Attrition in a 30-year follow-up of a perinatal birth risk cohort: factors change with age

Launes Jyrki 1 jyrki.launes@helsinki.fi
Hokkanen Laura 1
Laasonen Marja 1 2
Tuulio-Henriksson Annamari 1 3
Virta Maarit 1
Lipsanen Jari 1
Tienari Pentti J. 4 5
Michelsson Katarina 6
1 Faculty of Behavioral Sciences, Division of Cognitive and Neuropsychology, University of Helsinki , Helsinki , Finland
2 Helsinki University Central Hospital, Department of Phoniatrics , Helsinki , Finland
3 Kela—The Social Insurance Institution of Finland , Finland
4 Biomedicum, Research Programs Unit, Molecular Neurology, University of Helsinki , Finland
5 Helsinki University Central Hospital, Department of Neurology , Helsinki , Finland
6 Lecturer of pediatrics, retired
Niaura Raymond
Electronic publication date: 2014 Jul 8
Publication date: 2014
Volume: 2
Electronic Location ID: e480
Received 2014 Mar 31; Accepted 2014 Jun 20
Copyright: © 2014 Launes et al.
Copyright year: 2014
Copyright holder: Launes et al.
License: This is an open access article distributed under the terms of the Creative Commons Attribution License, which permits unrestricted use, distribution, and reproduction in any medium, provided the original author and source are credited.
License URL: https://creativecommons.org/licenses/by/3.0/

Keywords: Attrition, Longitudinal studies, Birth cohort, Cohort studies, Prospective, Follow-up, LTFU, Perinatal risks

Funding: No external funding for the present study was obtained.

==============================
Background. Attrition is a major cause of potential bias in longitudinal studies and clinical trials. Attrition rate above 20% raises concern of the reliability of the results. Few studies have looked at the factors behind attrition in follow-ups spanning decades.

Methods. We analyzed attrition and associated factors of a 30-year follow-up cohort of subjects who were born with perinatal risks for neurodevelopmental disorders. Attrition rates were calculated at different stages of follow-up and differences between responders and non-responders were tested. To find combinations of variables influencing attrition and investigate their relative importance at birth, 5, 9, 16 and 30 years of follow-up we used the random forest classification.

Results. Initial loss of potential participants was 13%. Attrition was 16% at five, 24% at nine, 35% at 16 and 46% at 30 years. The only group difference that emerged between responders and non-responders was in socioeconomic status (SES). The variables identified by random forest classification analysis were classified into Birth related, Development related and SES related. Variables from all these categories contributed to attrition, but SES related variables were less important than birth and development associated variables. Classification accuracy ranged between 0.74 and 0.96 depending on age.

Discussion. Lower SES is linked to attrition in many studies. Our results point to the importance of the growth and development related factors in a longitudinal study. Parents’ decisions to participate depend on the characteristics of the child. The same association was also seen when the child, now grown up, decided to participate at 30 years. In addition, birth related medical variables are associated with the attrition still at the age of 30. Our results using a data mining approach suggest that attrition in longitudinal studies is influenced by complex interactions of a multitude of variables, which are not necessarily evident using other multivariate techniques.

Introduction

Bias arises if the rate of subjects dropping out from a longitudinal study or clinical trial is not random, and the variables responsible for a subject to drop out from the study are correlated with variables that are used to evaluate outcome. Such bias is referred to as attrition bias. In epidemiology the term attrition is often used as a proxy for not retaining subjects who were initially included in a study. Thus, significant attrition may also occur without causing bias i.e., significant imbalance between study groups. However, other types of bias may be caused when subjects are lost to follow-up (LTFU) e.g., loss of statistical power when the groups get smaller. One of the dictionary definitions of attrition is to become weaker over the course of time and when used in conjunction with studies the term includes two sides of one problem i.e., the predictive power becomes weaker and the number of participants gets smaller. It is accepted that 5% of cases lost-to-follow-up is acceptable and attrition of up to 20 percent rarely presents serious bias (the 20% rule) (Fewtrell et al., 2008) In evidence based medicine LTFU of more than 20 percent downgrades a Level 1 study to a Level 2 study. The literature reports on a number of cohort studies and clinical trials, with attrition rates not disclosed. The 20% rule may be one reason for the authors’ reluctance to reveal high attrition rates (Fewtrell et al., 2008). Simulation studies suggest that the proposed 60–80% threshold of follow-up may cause significant bias especially when attrition is not completely random (Kristman, Manno & Côté, 2004). In order to interpret follow-up results, bias must be analyzed, corrected, and taken into account in clinical interpretation (Guyatt, 2009). However, even high proportions of LTFU do not necessarily cause uncontrollable bias for endpoint measures of a trial (Lacey, Jordan & Croft, 2013; Gustavson et al., 2012), e.g., a high proportion of LTFU makes it unreliable to calculate prevalences but cause—consequence analyses are still possible (Martikainen et al., 2007). In follow-up studies and trials that last for decades, it is next to impossible to achieve attrition less than 20–40%. Although bias is a major concern in the literature, surprisingly few attrition analyses have revealed substantial bias in estimated measures (Littman et al., 2010).

There are many characteristic features for individuals dropping out of studies, which are seen rather constantly: male gender, unmarried, smoking, low socioeconomic status (SES), and poor health (Goldberg et al., 2006; Young, Powers & Bell, 2006). Studies with aged subjects show a tendency of non-responders being older (Chatfield, Brayne & Matthews, 2005), but in studies with younger adults the opposite may be the case (Deeg, 2002; Young, Powers & Bell, 2006). Other sociodemographic factors linked to attrition include fewer years of education, ethnic group, no previous participation in research, and major depressive disorder (Lamers et al., 2012; Fröjd, Kaltiala-Heino & Marttunen, 2011). Finally, although the effect of personality of the subjects on attrition has not been extensively studied, personality, health, and lifestyle seem relatively neutral with respect to non-response (Distel et al., 2007).

Sometimes studies do not achieve good participation, even though previous experiences suggest otherwise (Korkeila et al., 2001). In one such study (Korkeila et al., 2001), the socioeconomical and behavioral features were important for non-response, but issues with privacy may have been decisive. A steady decrease of willingness to take part in studies has been observed in the western world in recent decades (Galea & Tracy, 2007). This has been attributed to profound changes of the day-to-day social and working environment, but especially to advances in communication technology that cause an increase in unsolicited contacts in form of aggressive telemarketing and campaigns cloaking as research studies. All these practices are thought to cause general mistrust to scientific studies in general, and cause different likelihood of participation, depending on the nature of the phenomenon studied. A potentially harmful environmental factor e.g., toxic chemicals to which a subject has been exposed may increase the probability of participating a study of that factor, but a potentially embarrassing life style habit (such as alcohol abuse or smoking) may discourage participation in a study of that habit.

Numerous practices have been suggested to overcome difficulties of attrition and enrolment. These include repeated mailings, telephone contacts, hybrid enquiries with the options of responding by either postal or internet response, and financial compensation. Face to face contact is associated with better participation rates in health promotion programs (Park et al., 2011). Perceived benefit of being involved counterbalances the degree of inconvenience caused by participating.

In studies involving children, the balance between wanting to participate and finding it inconvenient becomes complex, as both the parents’ and child’s motivation affect participation. This relationship also changes at different age levels. Intuitively it would seem natural that parents of sick children are motivated to participate because of more individualized follow-up and treatment. However, a meta-analysis of trials of children with chronic medical conditions, such as asthma, obesity, arthritis, diabetes, cancer, sickle cell disease, and cystic fibrosis showed a mean rate of enrollment refusal as high as 37% (range 0–75%), the initial mean follow-up attrition of 20% (range 0–54%), and extended follow-up attrition of 32% (range 0–59%) (Karlson & Rapoff, 2009). On the other hand, in the 26 year follow-up prospective cohort study in the United Kingdom on 1,064 children born small for gestational age, the participation on follow-up visits at 5, 10, 16, and 26 years was 93%, 80%, 72%, and 53%, respectively (Strauss, 2000). Up to the age of 10, the parents’ decision whether to participate or not has likely been decisive, at 16 the adolescents have had probably started to express their opinion and at 26 years, the now adult offspring have decided independently on participation. It is unlikely that the same causes of attrition apply at all phases of such long follow-up cohorts. There is no direct data about parents’ motives to respond, but in a 11-year longitudinal study of young females, frequent responders were likely to be more educated, less likely to be stressed about money, to smoke and to have children (Powers & Loxton, 2010). The importance of the child’s role can be evaluated from a study of parent–child interaction in families of preschoolers. Seventy-one percent of treatment dropouts were identified by lower SES, but poor child-parent-interaction was equally important (Fernandez & Eyberg, 2009). However, in cohort studies lasting for decades, withdrawing from a cohort study may also be temporary (Goldberg et al., 2006).

Clinical, psychological, and social features that could be identified at study entry influence attrition among adults. However, studies involving children are more complex in recruitment and retention. We report here attrition in a 30-year follow-up cohort of subjects who were born with a predefined perinatal risk for neurodevelopmental disorder. The aim is to analyze the causes of attrition, as well as to identify changes in the importance of factors associating to attrition at different stages of the follow-up. We hypothesize that factors contributing to attrition change due to the change of the person deciding on participation.

Material and Methods

Base cohort

The birth cohort originates from Kätilöopisto maternity hospital, Helsinki, Finland, in 1971–1974 and has been prospectively studied up to adulthood. The hospital is one of two major maternity hospitals, the other being the University Central Hospital, in the Helsinki metropolitan area of approximately 900 residents at the time of enrollment. The hospital served as the primary maternity hospital serving an area of approximately 4,000 km2. Sixty percent of the mothers lived in Helsinki (mean distance to hospital 8 km), 7% in Espoo (distance 15 km), 15% in Vantaa (distance 14 km) and the remaining 15% in the surrounding municipalities (average distance to hospital 30 km) (Fig. 1). All risk pregnancies known in advance as well as expected birth complications, were directed to the University Central Hospital equipped with a neonatal intensive care unit. Those born in Kätilöopisto therefore represent an unselected sample of deliveries within the area.

Figure 1 Area of residency.

The area of mothers’ place of residency. Dark (red): Helsinki, 60% of mothers (mean distance to hospital 8 km). Magenta: Espoo and Vantaa, 22% of mothers (mean distance 15 km). Pink: 18% of the mothers in the surrounding municipalities (average distance 30 km). (Adapted from the work of BishkekRocks under CC BY-SA 3.0, http://commons.wikimedia.org/wiki/File:Finnish_municipalities_2007.png.)

There were 22,359 consecutive births, which at the time accounted for about 10% of all births in Finland and about one third of those in the county of Uusimaa. Of those, 2,113 (9.5%) were cesarean sections. Newborns were consecutively screened for predefined perinatal risks: APGAR score lower than 7 at 5 or 15 min (n = 260), birth weight under 2,000 g or less (n = 221) out of whom 46 under 1500 g, significant hyperbilirubinemia (n = 353), severe respiratory difficulties including infections (n = 83), neurological symptoms (n = 114), maternal diabetes (n = 86), infant hypoglycemia (n = 92). Altogether 1,196 (5.3% of the 22,359 births) had at least one risk, 22% of them had more than one risk factor. Additionally, 130 cases were screened, but did not fulfill the inclusion criteria. Of the 1,196 infants, 202 died or were severely disabled and excluded from the study. The inclusion of patients is given in Fig. 2.

Figure 2 Inclusion.

Inclusion of subjects in the 30-year longitudinal follow-up study of children at risk for neurodevelopmental disorders.

Follow-up and other clinical visits

There were several clinical visits after birth, and the children were seen at 6, 12, 18, and 24 months by the physicians involved in the longitudinal study. Additionally, almost all infants were seen by a nurse at child-health centers as part of a public healthcare program, at 6, 10, 18, 24, and 48 months (Vuorenkoski, Mladovsky & Mossialos, 2008). Vaccinations were given on these visits following a schedule adapted from the WHO recommendations, and the vaccination program was continued in school health care. Therefore, the children and parents had frequent face-to-face contacts with health care personnel in addition to study visits. Moreover, clinical visits and psychological and logopedic test sessions took usually several hours, which means that the time parents were in contact with the study personnel was substantial. The parents did not receive financial compensation. The participants of the study had normal access to the public health care system at all times, as the study was not meant to substitute for public health services.

Methods

At birth, detailed data was recorded about maternal risk factors (e.g., smoking, prescribed medication and radiological examinations), family genetic traits, and medical data of delivery.

At five years, a clinical evaluation, age-appropriate psychological tests, logopedic tests, and questionnaires were used. The clinical evaluation included the Neurodevelopmental screen (NDS; Bax & Whitmore, 1973; Michelsson & Donner, 1981). The psychological evaluation included the Frostig Developmental Test of Visual Perception (Frostig, Lefever & Whittlesey, 1966), the Dubowizt screening test (Dubowitz, Leibowitz & Goldberg, 1977), and the Draw a Person Test (Goodenough, 1926). The logopedic evaluation included the Illinois Test of Psycholinguistic Abilities (Kirk, McCarthy & Kirk, 1968), a systematic evaluation of speech and articulation, and a Name writing test. The parents filled out a structured questionnaire regarding the health, development and behavior of the child. In addition, some kindergartens were contacted. The parents were also interviewed.

At nine years a clinical evaluation, age-appropriate psychological tests, logopedic tests, and questionnaires were again used. The clinical evaluation included the Test of Motor Impairment (Stott, Moyes & Henderson, 1972) and the assessment of neurological signs (Lindahl, Michelsson & Donner, 1988; Stokman et al., 1986). The psychological evaluation included the Draw a Person Test (Goodenough, 1926) and the Wechsler Intelligence Scale for Children (Wechsler, 1949). The logopedic evaluation included the Illinois Test of Psycholinguistic Abilities (Kirk, McCarthy & Kirk, 1968), the systematic evaluation of speech and articulation, the Diagnostic Reading and Writing tests for Finnish school children (Ruoppila, Roman & Vasti, 1968; Ruoppila, Roman & Vasti, 1969) and the Name Printing Test (Reimer et al., 1975). The parents filled out a structured questionnaire regarding the health, development and behavior of the child (Supplemental Information S1). Parents were asked about the usefulness of the study and rehabilitation measures initiated based on the visit at 5 years of age. The teachers filled out a structured questionnaire regarding school performance and behavior as well as the personality of the child (Supplemental Information S2). The parents were also interviewed face-to-face.

At 16 years of age, questionnaires were used for the whole cohort. A clinical evaluation and psychological tests, not analyzed in this paper, were conducted for a subsample of children. The clinical evaluation included the Test of Motor Impairment (Stott, Moyes & Henderson, 1972) and the assessment of neurological disorder (Lindahl, Michelsson & Donner, 1988; Stokman et al., 1986). The psychological evaluation included subtests from the Wechsler Adult Intelligence Scale (Wechsler, 1955). The questionnaires included early versions of the Child Behavior Check List for parents and the Youth Self-Report (Achenbach & Edelbrock, 1983; Achenbach & Edelbrock, 1987). The parents also filled out a structured evaluation of the child’s behavior and personality (Supplemental Information S3, S4, S5).

At 30 years, a questionnaire was used to explore educational and occupational outcome, health, and life satisfaction (Supplemental Information S6). It included also the ADHD Current Symptoms Scale as the ADHD Childhood Symptoms Scale with each of the 18 DSM-IV (American Psychiatric Association, 1994) diagnostic symptom criteria scored from zero to three depending on the severity of symptoms (Barkley & Murphy, 1998).

The family’s socioeconomic status (SES) was recorded at all levels. The social class variable was defined as the best score on a 5-item scale based on father’s occupation, and if unavailable, mother’s occupation was used. The psychosocial distress score was formed to include poor housing conditions, divorces, relocations, alcohol abuse, unemployment, family conflicts, domestic violence, imprisonment, severe diseases, or mental problems in the family.

Statistics

Participation frequencies were calculated at birth, 5, 9, 16 and 30 years. The children, who were not seen on the 5-year visit or any other visit thereafter, were included only in the analysis of 5-year results. First, we tested for significant group differences with the Mann–Whitney-U test with participation at different age levels as the grouping variable. Ordinal and categorical variables were cross-tabulated, the Fisher’s exact test or the Pearson Chi-square test were used for significance testing and Cramér’s V for association. Regression methods were not used, because in our follow-up cohort, neither the dependent nor the predictor variables belong to the same pool of variance as the person who decides of participation changes in course of the follow-up. Bonferroni correction was used to correct the p values for the required alpha level of 0.05 due to multiple significance testing. Statistica 12 software was used for the calculations (Statsoft, 2013).

To identify clinically meaningful patterns and combinations of explanatory variables, we performed a classification tree analysis using the random forest algorithm (Breiman, 2001a). In a situation with a very large numbers of predictors which are likely to interact in an unknown fashion, classification tree analysis (e.g., Rokach and Maimon, 2007) provides a flexible and robust way to find the most important predictors of attrition. The present data was collected at fixed time points. Therefore, classification tree analysis was used instead of survival analysis, which is often used to analyze study designs where time is measured as a continuous variable. The basic idea of classification tree analysis is simple. Prediction of a response or class Y from inputs X1, X2, …, Xp is done by growing a binary tree. At each internal node in the tree, one applies a test to one of the inputs Xi. Depending on the outcome of the test, one goes to either the left or the right sub-branch of the tree. Eventually one comes to a leaf node, where prediction is made. This prediction aggregates or averages all the training data points that reach that leaf. In contrast to this, predictors like linear or polynomial regression are global models, where a single predictive formula is supposed to hold over the entire data space. As noted previously, when the data has lots of features which interact in complicated, and possible nonlinear ways, assembling a single global model can be very difficult. Detailed presentation of classification tree algorithms can be found e.g., in Breiman et al. (1984) and in Kohavi & Quinlan (2002).

Data mining techniques are useful in exploratory studies (Mendez et al., 2008) when data cannot be fit into models using goodness-of-fit methods and statistically significantly differences are difficult to interpret. As compared to such classical data mining situations the sample size in the present study is small. Therefore the random forest algorithm (Breiman, 2001a) was used, because it prevents overfitting the model which is problem in classical classification tree analysis (Breiman, 2001b). Model accuracy was optimized by testing predictive accuracy, which was in the range of 0.71–0.97 at different age levels. The Gini index (Gini, 1909) was used as a measure for variable importance. Gini index is the weighted mean of the improvement that each variable causes for the splitting criterion. Maximal classification accuracy and the most important variables using Gini importance score are reported (Sandri & Zuccolotto, 2006). Random forest classification was done using the Salford system Predictive Modeler software (Salford Systems, 2013). In all analysis Bootstrap aggregating was used for training algorithm (Breiman, 1996). Detailed nontechnical description of random forest algorithm can be found in Touw et al. (2013).

Ethics

Ethical review has been conducted over the course of the longitudinal study, and the latest approval was obtained from the Ethical Review Board of the Helsinki and Uusimaa hospital district in May 2013 (number 147/13/3/00/2013).

Results

Attrition

There were 130 infants who were seen neither at the 5-year visit nor on any later visit. The initial loss during the first 5 years therefore was 13%. However, 92 of them had been seen at an early clinical visit or child health centers during the first 40 months of life, thus only 38 were completely LTFU after birth. At 5 years, 839 children were assessed (84% of those alive and not severely disabled, n = 994), at the age of 9 years 756 (76%), including 25 children not assessed at 5 years. At the age of 16 years 560 responded (65% of those who participated at either 5 or 9 years, n = 864) and at the age of 30 years 469 responded (54%). The attrition rate thus increased from 16% at 5 years to 46 at 30 years. There was data on 78 (8%) children on one age level only, on 152 (15%) children on two levels, on 288 (29%) children on three levels, and on 343 (35%) on all four age levels of the study. Participation of subjects on the different age levels is given in Figs. 3 and 4. At the age of 9 years 54% of the parents regarded participation in the study useful, 26% were undetermined, 13% did not answer that question, and 7% did not find participation useful.

Figure 3 Participation.

Participation of subjects in the 30-year longitudinal follow-up study of children in risk of neurodevelopmental disorders. Cases excluded due to death or severe handicap are not shown. Total number of potential participants was 994 at 5 and 9 years, and 864 at 16 and 30 years. Attrition thus was 16% at five, 24% at nine, 35% at 16 and 46% at 30 years. Numbers on the left column indicate participation at different age levels.

Figure 4 Retention rate.

The retention rate of the follow-up of a cohort of 1,196 neonates with birth risks. The 202 children who died or were severely handicapped were excluded, thus the actual number of included children was 994 (dotted line). * Only the children who had participated at 5 or 9 years were invited at 16 years and 30 years.

Group comparisons of responders and non-responders

Between the groups of children who were not brought to clinical evaluations at all (n = 130) and those who did not participate at five years, but participated later (n = 25), there was a statistically significant difference only in the father’s social class (mean responders 2.2 ± 0.9 and 2.6 ± 1.0 in non-responders). The social class was the only variable that was significantly different between responders and non-responders at all other age levels too, a lower socioeconomic class being associated with non-responding.

A complete list of 63 variables tested for group differences is presented in Supplemental Information S7. Notably, the type of perinatal risk had no apparent effect on the number of visits at any age level (Kruskal–Wallis test H = 8.10, p = 0.32). Also, maternal age, gestational age, birth weight, or Apgar scores or the distance from home to the hospital had no statistically significant differences on any age level.

Classification analysis

Classification accuracy of classifying cases into responders and non-responders, and five of the most important variables with their relative importance are given in Table 1. A complete list of variable importance is available in Supplemental Information S8. The variables that got a high importance score in the random forest model were grouped into three categories: 1. a category characterizing the mother, infant and delivery (Birth related), 2. a category characterizing the neurodevelopmental factors and body mass index (Development related), and 3. a category characterizing the socioeconomic status, perceived security of the child and living conditions of the family (SES related).

Table 1 Classification accuracy and five of the most important variables at each age level selected for classification by the random forest classification model.

Accuracy (overall fraction correct) calculated from 2∗2 contingency table (a + d/t).

	At birth	Age 5	Age 9	Age 16	Age 30	
Birth related	Birth weight	1.00	Mother’s age	1.00	Gestational weeks	0.52	Mother’s age	0.56	Mother’s age	0.87	
	Mother’s age	0.78	Birth weight	0.69	Birth weight	0.48	Birth weight	0.37	Pregnancy complication	0.58	
	Resuscitation	0.77	Hyperbilirubinemia	0.58	Mother’s age	0.44	Pregnancy	0.31	Birth weight	0.41	
	Perinatal treatments	0.75	Pregnancy	0.56	Hyperbilirubinemia	0.40	Apgar 1 min	0.20	X-ray in pregnancy	0.26	
	Gestational weeks	0.75	Gestational weeks	0.55	Respiratory	0.38	Apgar 5 min	0.16	Apgar 1 min	0.22	
SES related	Distance	0.55	Distance	0.41	Housing (5)	0.85	Childs’ security (9)	0.44	Parents’ view (16)	0.79	
	Father’s social class	0.54	Mother working	0.39	Parity (5)	0.80	Housing (5)	0.33	Childs’ security (9)	0.66	
	Mother working	0.44	Father’s social class	0.36	Domestic dispute (5)	0.53	Special education (9)	0.30	Special education (9)	0.37	
	Smoking in pregnancy	0.35	Smoking in pregnancy	0.31	No of occupants (5)	0.45	Father’s social class	0.19	Current activity (16)	0.35	
	Marital status	0.29	Marital status	0.23	Father’s social class	0.45	Mother working	0.14	Subjective gain (9)	0.30	
Neurodevelopmental related					Dubowitz test (5)	1.00	NDS (5)	1.00	NDS (5)	1.00	
					DAP (5)	0.90	DAP (9)	0.38	Dubowitz test (5)	0.54	
					NDS coordination (5)	0.72	WISC (9)	0.37	ITPA (5)	0.53	
					ITPA (5)	0.72	TOMI (9)	0.33	ITPA (9)	0.48	
					NDS (5)	0.71	ITPA (9)	0.32	WISC VIQ (9)	0.48	
Classification accuracy	0.81		0.74		0.90		0.76		0.96		
(95% confidence limits)	0.78–0.82		0.71–0.76		0.87–0.91		0.72–0.79		0.94–0.97		
Notes.

Numbers in parenthesis refer to the age in which the variable is measured.

BMI Body mass index

DAP Draw a Person test

ITPA Illinois test of psycholinguistic ability

TOMI Test of motor impairment

NDS Neurodevelopmental screen

WISC Wechsler intelligence test for children

VIQ Verbal intelligence quotient

PIQ Performance intelligence quotient

For assessing the response of not participating at all, the average importance of the variables selected by the random forest model was 0.50 (range 1.0–0.16; 21 variables) in the Birth related category and 0.39 (range 0.55–0.17; six variables) in the SES related category. At five years the average importance was 0.39 (range 1.0–0.13; 24 variables) in the Birth related category, and 0.27 (range 0.41–0.08; seven variables) in the SES related category. At nine years the average importance was 0.28 (range 0.52–0.09; 20 variables) in the Birth related category, 0.39 (range 0.85–0.07; 13 variables) in the SES related category, and 0.81 (range 1.00–0.71; five variables) in the Development related category. At 16 years of age the average importance of variables in the Birth related category was 0.19 (range 0.44–0.03; 11 variables), 0.14 (range 0.56–0. 03; 14 variables) in the SES related category and 0.36 (range 1.00–0.23; 12 variables) in the Development related category. At 30 years of age the corresponding figures were 0.31 (range 0.87–0.10; 11 variables) for in Birth related category, 0.25 (range 0.79–0.06; 17 variables) in the SES related category and 0.50 (range 1.00–0.39; 13 variables) in the Development related category (Fig. 5).

Figure 5 Variable importance in random forest model.

Average importance of variables which predict participation/non-participation as calculated by the random forest classification model. The variables were classified into three categories which are (1) variables relating to child, delivery and pregnancy (Birth), (2) variables relating to neurodevelopment and behavior (Development), and (3) variables reflecting socioeconomic status (SES). Columns represent each age level of the longitudinal follow-up and are scaled to 100%.

Within the Birth related category, at all age levels, maternal age, gestational age, birth weight, and Apgar scores were in the upper quartile of the variable importance scores.

Within the SES related category, up to the age of 16, the most important variables the model chose for the classification into responders and non-responders reflected the family’s occupation, housing, domestic disputes, and distance to hospital. At 30 years, the most important variables reflected more the individual’s characteristics, e.g., subjective feeling of security, plans for future, parent’s opinions regarding the child and to a lesser degree the socioeconomic features of the family in which the individual was raised.

Within the Development related category, the Neurodevelopmental screen was the most important variable for classifying into responders and non-responders at 16 and 30 years and it was the fifth most important variable at nine years.

Discussion

The aim of the study was to analyze the causes of attrition, as well as to identify changes in the importance of factors associating to attrition at different stages of a 30-year follow-up of a cohort born with perinatal risks. Initial loss of potential participants was 13%. Attrition was 16% at five, 24% at nine, 35% at 16, and 46% at 30 years. There were very few differences in group comparisons of responders and non-responders, when traditional significance tests were used. The only finding was a lower socioeconomic class being associated with non-responding at all age levels. Using the random forest model the variables were grouped into Birth related, Development related, and SES related categories. Birth related variables were important predictors of responding pattern during the first 5 years of the follow-up. The Development related variables became highly important at 9 years and appeared important at 30 years of age. The importance of SES related variables remained stable all through the follow-up but compared to the other categories, did not exceed their importance at any time point.

Measuring the relative importance of variables affecting attrition in very long lasting longitudinal studies, or even demonstrating which variables have any effect at all, is difficult using traditional stochastic models such as logistic regression. Logistic regression assumes that the conditional probabilities are a logistic function of the independent variables, which cannot be true when predictors may originate from either the parent or the child. Data mining techniques can be used instead when data cannot be fit into regular models (Mendez et al., 2008). In this analysis, as group level differences were negligible, we felt it unnecessary to correct for covariates using multivariate methods. On the other hand, using the random forest classifier, a very accurate (up to 96%) classification of cases as responders and non-responders was achieved. The classification target variable was participation or nonparticipation at birth, 5, 9, 16, and 30 years in the random forest analysis. It seemed reasonable to assume that factors affecting the attrition may be “inherited” over age levels. Thus, variables from all previous levels were included in the classification analysis, e.g., data from the risks and events of the pregnancy and delivery as well as data from 5 years were used as predictors for classification at 9, 16, and 30 years. This allows for an estimation of both the quality and the quantity of the most important factors, as well as how they change over time (Breiman, 2001a; Breiman, 2001b; Genuer, Poggi & Tuleau-Malot, 2010).

At the age of nine, 16 and 30 years, variables in all of the three categories, i.e., birth, SES and development related categories, influenced the classification. On all age levels after 5 years, the variables belonging to the development related category had the most influence (Fig. 5, Table 1) in the random forest model. The variables in this category are scores of cognitive tests, parents’, and teachers’ assessments, behavior and motor skills, as well academic achievement, e.g., school grades. Such properties have not been reported as attrition factors in studies with younger subjects, but in the elderly, e.g., cognitive status is an important factor (Chatfield, Brayne & Matthews, 2005).

As commonly observed in studies (Fewtrell et al., 2008), the important SES related variables included area of residency, low social class, marital status, smoking and mothers working during pregnancy. However, there were no statistically significant group differences other than SES level. The distance to hospital was among the most important variables in the classification analysis up to the age of five, possibly because traveling with small children requires more time and attention. There appears to be a qualitative change in the most important SES related variables between 16 and 30 years, and the subjective benefit from participation was not among the important variables until the age of 30.

At every age level the birth related variables, maternal age, gestational age, birth weight, and Apgar scores, were in the upper quartile of the variable importance scores. The medical events during pregnancy and in delivery had substantial influence in classifying between responders and non-responders on first clinical visits. The effect was still evident at 30 years, despite the fact that at this age the decision to respond was made by the subjects themselves instead of their parents. So far, it has not been reported that birth related factors are “inherited” as classifiers of response pattern to adult children. Repeatedly measured variables are inter-correlated by nature, but owing to the random forest algorithm’s capability to handle missing data and correlated variables, and good resistance to overfitting, we think that this finding is not a mathematical artifact.

The magnitude to which the parents’ attitudes are transferred to progeny by social inheritance of attitudes and negative dispositions towards volunteer activities and research (Park et al., 2011) cannot be analyzed in our material. On the other hand, the frequent face-to-face contacts may have improved inclusion and staying in the study. In addition to the study visits, our cohort had frequent scheduled contacts with both the public health care system (programs of child health centers, school health care). Such frequent contacts are generally considered a motivational factor for parents. In Finland, there is also clear social pressure in keeping children in these programs, which is exemplified by a high success rate of vaccination programs running simultaneously (Peltola et al., 1994). We regard it highly probable that these frequent visits have reduced over-all attrition, as more than one half of the parents regarded participation in the study as useful and only 7% not useful at all. However, as there were differences in the social class of the families and SES variables were important throughout the follow-up, frequent face-to-face contacts may not have the desired effect in reducing bias.

We conclude that attrition in our cohort is well within the same range as other lengthy studies; especially the nonresponse rate in young adulthood is strikingly similar to the 26-year follow-up of children born small for gestational age (Strauss, 2000). Based on other literature, many of the longitudinal cohorts have not been badly influenced by attrition bias. We found very few variables that were significantly different in group comparisons of responders and non-responders. However, the lack of statistically significant differences does not prove the absence of clinically relevant difference. Using a classification analysis, we found a number of variables that very accurately predicted response or nonresponse. These variables could be classified into three main dimensions describing events of pregnancy and birth, growth and neurodevelopment, as well as socioeconomic status. By performing this analysis, we aimed to demonstrate a much more complex network of dependencies that has been described in previous studies. The most important difference between previously reported studies and our analysis is methodological. Using traditional statistical tools, we found only one foreseeable difference, father’s occupationally defined social group. Therefore, we are confident, that future subgroup analyses of our cohort will be reasonably unbiased as other cohorts resembling ours. Analyses based on data mining techniques seem to reveal unexpected interactions, but our method must be confirmed by similar analyses.

Supplemental Information

Supplemental Information S1 Background information at 9 years

Click here for additional data file.

Supplemental Information S2 Teacher’s assessment questions

Click here for additional data file.

Supplemental Information S3 Background information at 16 years

Participation of subjects in the 30-year longitudinal follow-up study of children in risk of neurodevelopmental disorders. Cases excluded due to death or severe handicap are not shown. Total number of potential participants was 994 at 5 and 9 years, and 864 at 16 and 30 years. Attrition thus was 16% at five, 24% at nine, 35% at 16 and 46% at 30 years. Numbers on the left column indicate participation at different age levels.

Click here for additional data file.

Supplemental Information S4 Questionnaire to parents at 16 years

Click here for additional data file.

Supplemental Information S5 Questionnaire at 30 years

English translation of the questionnaire form used at 30 years.

Click here for additional data file.

Supplemental Information S6 Plans at 16 years

Questionnaire of subjects’ plans for future at 16 years.

Click here for additional data file.

Supplemental Information S7 Variables, tested for statistically significant group differences for response or nonresponse at birth (postnatally) and the ages of 5 years, 9 years, 16 years, and 30 years

Click here for additional data file.

Supplemental Information S8 Variable importance, all variables

Variable importance given by random forest classification model, all variables included.

Click here for additional data file.

Additional Information and Declarations

Competing Interests

Author Contributions

Human Ethics

The authors declare there are no competing interests.

Jyrki Launes conceived and designed the experiments, analyzed the data, wrote the paper, prepared figures and/or tables.

Laura Hokkanen and Katarina Michelsson conceived and designed the experiments, analyzed the data, wrote the paper, prepared figures and/or tables, reviewed drafts of the paper.

Marja Laasonen, Annamari Tuulio-Henriksson, Maarit Virta and Pentti J. Tienari conceived and designed the experiments, wrote the paper, reviewed drafts of the paper.

Jari Lipsanen conceived and designed the experiments, analyzed the data, wrote the paper, reviewed drafts of the paper.

The following information was supplied relating to ethical approvals (i.e., approving body and any reference numbers):

Ethical review has been conducted over the course of the longitudinal study, and the latest approval was obtained from the Ethical Review Board of the Helsinki and Uusimaa hospital district in May 2013 (number 147/13/3/00/2013).

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
