# Peer review of "Attrition in a 30-year follow-up of a perinatal birth risk cohort: factors change with age"

_PeerJ, doi:10.7717/peerj.480_

## Round 0.1 · original submission · Major Revisions

In your revision, please be sure to address every point raised by the reviewers, whether you agree or not.

·

Basic reporting

No comments

Experimental design

no comments

Validity of the findings

No comments

Additional comments

The paper is interesting and very relevant and I have enjoyed to read it

·

Basic reporting

The study is carefully carried out, and the manuscript deals with an important aspect that concerns all long-term follow-up studies.
The aims are 1) to describe attrition as a general phenomenon, 2) to analyze the causes of attrition and to find out whether those causes are different according to the years of the follow-up and finally 3) to use one’s own study setting as an example of the attrition.

Experimental design

The design is correct and important.

Validity of the findings

The authors demontrate in their data the concept of attrition. The attrition is a frequent source of bias in epidemiological studies and the analysis of attrition is essential.

Additional comments

I have few major concerns and a couple of stylistic remarks.
First of all, attrition as a concept has to be defined, and also its connections to the concepts of lost-to-follow-up, retention rate, and drop-out (analyses). I realize that attrition is a polite name for the phenomenon caused by drop-outs, but the concept is rarely used so it is good to describe it appropriately.
In the method section, the authors should better describe the random forest model and the data mining procedure, because it may be quite unknown for most of the readers who are more familiar using the survival analysis (Cox regression model). Survival analysis allows the use of different explanatory variables in the model, but it may not be a reliable method in this case where the predictors can originate from either the parent or the child, as the authors explain.
Moreover, the authors should add a figure about the total retention rate. I enclose here an example.


It is often said that the legends of the figures and tables should be so well constructed that they should be able to stand alone, separate from the text of the article, and be understood. The legends in this manuscript need better explanation, especially figure 4.
With these alterations, it is much easier to follow the text and the figures and tables.
In the introduction, it would be good to add some explanations after the sentence beginning in the line 26. E.g. High proportion of LTFU makes it unreliable to calculate prevalences but cause-consequence analyses are often still possible (e.g.Martikainen P, Laaksonen M, Piha K, Lallukka T. Does survey non-response bias the association between occupational social class and health? Scand J Public Health 2007; 35:212-5)

Some minor aspects:
- line 118: as part (without article)
- a sentence in the line 125 is a bit difficult, clarify
- line 157: the assessment of neurological disorder

---

## Round 0.2 · accepted · Accept

Thank you for your submission to PeerJ. I am writing to inform you that your manuscript, "Attrition in a 30-year follow-up of a perinatal birth risk cohort: factors change with age"